# Dimensionality, Granularity, and Differential Residual Weighted Entropy

**DOI:** 10.3390/e21090825

**Published:** 2019-08-23

**Authors:** Martin Tunnicliffe, Gordon Hunter

**Affiliations:** School of Computer Science and Mathematics, Kingston University, Penrhyn Road, Kingston-on-Thames KT1 2EE, UK

**Keywords:** weighted entropy, residual entropy, differential entropy, dimensionality

## Abstract

While Shannon’s differential entropy adequately quantifies a dimensioned random variable’s information deficit under a given measurement system, the same cannot be said of differential weighted entropy in its existing formulation. We develop weighted and residual weighted entropies of a dimensioned quantity from their discrete summation origins, exploring the relationship between their absolute and differential forms, and thus derive a “differentialized” absolute entropy based on a chosen “working granularity” consistent with Buckingham’s Π-theorem. We apply this formulation to three common continuous distributions: exponential, Gaussian, and gamma and consider policies for optimizing the working granularity.

## 1. Introduction

Informational entropy, introduced by Shannon [1] as an analogue of the thermodynamic concept developed by Boltzmann and Gibbs [2], represents the expected information deficit prior to an outcome (or message) selected from a set or range of possibilities with known probabilities. Many modern applications using this concept have been developed, such as the so-called maximum entropy method for choosing the “best yet simplest” probabilistic model from amongst a set of parameterized models, which is statistically consistent with observed data. Informally, this can be stated as: “In order to produce a model which is statistically consistent with the observed results, model all that is known and assume nothing about that which is unknown. Given a collection of facts or observations, choose a model which is consistent with all these facts and observations, but otherwise make the model as ‘uniform’ as possible” [3,4]. Philosophically, this can be regarded as a quantitative version of “Occam’s razor” from the 14th Century - “Entities should not be multiplied without necessity”. Mathematically, this means that we find the parameter values which maximize the entropy of the model, subject to constraints that ensure the model is consistent with the observed data, and MacKay [5] has given a Bayesian probabilistic explanation for the basis of Occam’s razor. This maximum entropy approach has found widespread applications in image processing to reconstruct images from noisy data [6,7] - for example, in Astronomy, where signal to noise levels are often extremely low - and in speech and language processing, including automatic speech recognition and automated translation [3,8].

This idea of informational entropy has been expanded and generalized. Tsallis [9] proposed alternative definitions to embrace inter-message correlation [10], though the information of a potential event remained solely dependent on its unexpectedness or “surprisal”. This is somewhat counterintuitive: “Man tosses 100 consecutive heads with coin” is very surprising but not important enough to justify a front-page headline. Conversely “Sugar rots your teeth” is of great importance but its lack of surprisal disqualifies it as news. “Aliens land in Trafalgar Square” is both surprising and important and we would expect it be a lead story. To reflect this, Guiaşu [11] introduced the concept of “weighted entropy” whereby each possible outcome carried a specific informational importance, an idea expanded by Taneja and Tuteja [12], Di Crescenzo and Longobardi [13], and several others. Another modification considers the entropy of outcomes subject to specific constraints: for example, “residual entropy” was defined by Ebrahimi [14] for lifetime distributions of components surviving beyond a certain minimum interval.

From the outset, Shannon identified two kinds of entropy: the “absolute” entropy of an outcome selected from amongst a set of discrete possibilities [1] (p. 12) and the “differential” entropy of a continuous random variable [1] (p. 36). The differential version of weighted entropy has found several applications: Pirmoradian et al. [15] used it as a quality metric for unoccupied channels in a cognitive radio network and Tsui [16] showed how it can characterize scattering in ultrasound detection. However, under Shannon’s definition the differential entropy of a physical variable requires the logarithm of a dimensioned quantity, an operation which necessitates careful interpretation [17].

In this paper we examine the implications of this dimensioned logarithm argument to weighted entropy and show how an arbitrary choice of unit system can have profound effects on the nature of the results. We further propose and evaluate a potential remedy for this problem; namely a finite working granularity.

## 2. Absolute and Differential Entropies 

Entropy may be regarded as the expected information gained by sampling a random variable (RV) with a known probability distribution. For example, if X is a discrete RV and $p_{X}\left(x\right)=\mathrm{Pr}\left(X=x\right)$ then outcome X=x occurs on average once every $1/p_{X}\left(x\right)$ observations and the mean information encoded as $\log_{2}1/p_{X}=-\log_{2}p_{X}$ bits. However, it is common to use natural logarithms for which the information unit is the “nat” (≈1.44 bits). Entropy can therefore be defined as
(1)$$H\left(X\right)=E\left[-\log p_{X}\left(X\right)\right]=-\underset{x\in \Omega }{\sum }p_{X}\left(x\right)\log p_{X}\left(x\right)$$
where Ω is the set of all possible X. (An implicit assumption is that X results from an independent identically distributed process: while Tsallis proposed a more generalized form to embrace inter-sample correlation [9,10], the current paper assumes independent probabilities.) Shannon extended (1) to cover continuous RVs as “differential” entropy
(2)$$h\left(X\right)=E\left[-\log f_{X}\left(X\right)\right]=-\int_{-\infty }^{\infty }f_{X}\left(x\right)\log f_{X}\left(x\right)dx$$
where $f_{X}\left(x\right)$ is the probability density function (PDF) of X. Two points may be noted: firstly, since in (2) *x* only affects the integrand through $f_{X}\left(x\right)$, h(X) is “position-independent”, i.e., h(X)=h(X+b) for all real b. Secondly h(X) is not, as one might naïvely suppose, the limit of H(X) as resolution tends to zero (see Theorem 9.3.1 in [18]). Furthermore, while H(X) is always positive (since $0\leq p_{X}\left(x\right)\leq 1)$, h(X) may be negative if most larger values of $f_{X}\left(x\right)$ are ≥1. In the extreme case of a Dirac delta-function PDF, representing a deterministic—and therefore non-informative—outcome, the differential entropy would not be zero but minus infinity.

Take for example the Johnson-Nyquist noise in an electrical resistor: if the noise potential $v_{n}$ is Gaussian with an RMS value ε volts it is easy to show that $h\left(v_{n}\right)=\log\sqrt{2\pi }+\log\varepsilon +1/2$ nats. (Position-independence makes the bias voltage irrelevant.) Suppose that ε=0.4μV; working in microvolts we obtain $h\left(v_{n}\right)=0.5026$ nats but in millivolts $h\left(v_{n}\right)=-6.405$ nats. If differential entropy truly represented information then a noise sample in microvolts would increase our information but measured in millivolts would decrease it. Thus, h(X) must be regarded as a relative, not an absolute measure and consistent units must be used for different variables to be meaningfully compared.

“Residual” entropy, where only outcomes above some threshold t are considered, is given by [14]
(3)$$h\left(X;t\right)=E\left[-\log\frac{f_{X}\left(X\right)}{{\bar{F}}_{X}\left(t\right)}|X\geq t\right]=-\int_{t}^{\infty }\frac{f_{X}\left(x\right)}{{\bar{F}}_{X}\left(t\right)}\log\frac{f_{X}\left(x\right)}{{\bar{F}}_{X}\left(t\right)}dx$$
where ${\bar{F}}_{X}\left(t\right)=\int_{t}^{\infty }f_{X}\left(x\right)dx$ is called the “survival function” since in a life-test experiment it represents the proportion of the original component population expected to survive up to time t. Some authors call $f_{X}\left(x\right)/{\bar{F}}_{X}\left(t\right)$ the “hazard function” (a life-test metric equal to failure rate divided by surviving population) though this is only valid for the case of x=t; it is better interpreted as the PDF of X subject to the condition X≥t. This somewhat eliminates the positional independence since a shift in X only produces the same entropy when accompanied by an equal shift in t, i.e., h(X;t)=h(X+b;t+b), but the contribution of each outcome to the total entropy still depends on rarity alone.

Guiaşu’s aforementioned weighted entropy [11] introduces an importance “weighting” w(x) to outcome X=x whose surprisal remains $-\log p_{X}\left(x\right)$: the overall information of this outcome is redefined $w\left(x\right)\times -\log p_{X}\left(x\right)$ so entropy becomes $H^{w}\left(X\right)=-\underset{x\in \Omega }{\sum }w\left(x\right)p_{X}\left(x\right)\log p_{X}\left(x\right)$. It seems intuitively reasonable that the differential analogue should be $-\int_{-\infty }^{\infty }w\left(x\right)f_{X}\left(x\right)\log f_{X}\left(x\right)dx,$ though if w(x) is a monotonic function we could define this more compactly as [13]:(4)$$h^{w}\left(X\right)=E\left[-X\log f_{X}\left(X\right)\right]=-\int_{-\infty }^{\infty }xf_{X}\left(x\right)\log f_{X}\left(x\right)dx$$ and the residual weighted entropy
(5)$$h^{w}\left(X;t\right)=E\left[-X\log\frac{f_{X}\left(X\right)}{{\bar{F}}_{X}\left(t\right)}|X\geq t\right]=-\int_{t}^{\infty }x\frac{f_{X}\left(x\right)}{{\bar{F}}_{X}\left(t\right)}\log\frac{f_{X}\left(x\right)}{{\bar{F}}_{X}\left(t\right)}dx.$$

We have already noted that the logarithms of probability densities behave very differently from those of actual probabilities. Aside from the fact that $f_{X}\left(x\right)$ may be greater than 1 (a negative entropy contribution) it is also typically a dimensioned quantity: for example if x represents survival time then $f_{X}\left(x\right)$ has dimension [Time]^−1^, leading to the importance of unit-consistency already noted. In the next section we explore more deeply the consequences of dimensionality.

## 3. Dimensionality

The underlying principle of dimensional analysis, sometimes called the “Π-theorem”, was published in 1914 by Buckingham [19] and consolidated by Bridgman in 1922 [20]. In Bridgeman’s paraphrase [20] (p. 37) an equation is “complete” if it retains the same form when the size of the fundamental units is changed. Newton’s Second Law for example states that F=ma where F is the inertial force, m the mass and a the acceleration: if in SI units m=2 kg and a=2 ms^−2^ then the resulting force F=2×2=4N, where the newton N is the SI unit of force. In the CGS system m=2000 g and a=200 cms^−2^ so the force is 2000×200=400,000 dynes, the exact equivalent of four newtons. The equation is therefore “complete” under the Π-theorem which requires that each term be expressible as a product of powers of the base units: in this case [Mass][Length][Time]^−2^.

The problem of equations including logarithms (and indeed all transcendental functions) of dimensioned quantities has long been recognized. Buckingham opined that “… no purely arithmetic operator, except a simple numerical multiplier, can be applied to an operand which is not a dimensionless number, because we cannot assign any definite meaning to the result of such an operation” ([19], p. 346). Bridgman was less dogmatic, citing as a counter-example the thermodynamic formula $\lambda =RT^{3}\frac{d\log p}{dT}$ where T is the absolute temperature, p is pressure, and R and λ are other dimensioned quantities ([20], p. 75). It is true that the logarithm returns the index to which the base (e.g., e= 2.718…) must be raised in order to obtain the argument: for example if p=200 Pa (the Pa or pascal being the SI unit of pressure) then to what index must p be raised to in order to obtain that value? It is not simply a matter of obtaining 200 from the exponentiation but 200 *pascals*. Furthermore, the problem would change if we were to switch from SI to CGS where the pressure is 2000 barye (1 barye being 1 dyne cm^−2^) though the physical reality behind the numbers would be the same.

However, in the current case it is the derivative of log pressure which is important, and since $\frac{d\log p}{dT}=\frac{1}{p}\frac{dp}{dt}$ it has dimension [Temperature]^−1^ and the Π-theorem is therefore satisfied. Unfortunately, Shannon’s differential entropy $h\left(X\right)=-\int_{-\infty }^{\infty }f_{X}\left(x\right)\log f_{X}\left(x\right)dx$ has no such resolution since it is the absolute value of $f_{X}\left(x\right)$ (not merely its derivative) which must have a numeric value. This kind of expression has historically provoked much debate and though there are several shades of opinion we confine ourselves to two competing perspectives:

Molyneux [21] maintains that if m=10 grams then logm should be correctly interpreted as log(10×gram)=log(10)+log(gram) and “log(gram)” should be regarded as a kind of “additive dimension” (he suggests the notation 2.303 <gram>).

Matta et al. [17] argue that “log(gram)” has no physical meaning; while Molyneux had dismissed this as pragmatically unimportant, they echo the views of Buckingham [19] saying that dimensions are “… not carried at all in a logarithmic function”. According to Matta, logm must be interpreted as log(m/gram) (the dimension of m cancelled out by the unit).

Since most opinions fall more or less into one or other of these camps it will be sufficient to consider a simple dichotomy: we refer to the first of these as “Molyneux” and the second as “Matta”. Under the Molyneux interpretation the differential entropy must be expressed
(6)$$h\left(X\right)=-\int_{-\infty }^{\infty }f_{X}\left(x\right)\log f_{X}\left(x\cdot \mathrm{second}\right)dx=-\int_{-\infty }^{\infty }f_{X}\left(x\right)\log f_{X}\left(x\right)dx+\log\left(\mathrm{second}\right)$$
which has an additive (and physical) dimension of “log(second)” (or <second>) in addition to the multiplicative (and non-physical) dimension of nats. Pragmatically this is not important since entropies of variables governed by different probability distributions may still be directly compared (assuming X is always quantified in the same units). However, when we consider weighted entropy, we find that
(7)$$\begin{array}{ll} h^{w}\left(X\right) & =-\int_{-\infty }^{\infty }xf_{X}\left(x\right)\log f_{X}\left(x\cdot \mathrm{second}\right)dx\\ & =-\int_{-\infty }^{\infty }xf_{X}\left(x\right)\log f_{X}\left(x\right)dx+E\left[X\right]\log\left(\mathrm{second}\right) \end{array}$$
where E[X] is the expectation of X. Here Molyneux’s approach collapses since the expression has a multiplicative dimension nat-seconds and an additive dimension “E[X]log(second)”. Since the latter depends on the specific distribution, $h^{w}\left(X\right)$ loses any independent meaning; comparing weighted entropies of two different variables would be like comparing the heights of two mountains in feet, defining a foot as 12 inches when measuring Everest and 6 when measuring Kilimanjaro.

So, if Molyneux’s interpretation fails, does Matta’s fare any better? Since Matta requires the elimination of dimensional units, we introduce the symbol $\Delta_{X}$ to represent one dimensioned unit of X (for example, if X represents time in seconds then $\Delta_{X}=1$ s). The Shannon differential entropy now becomes $h\left(X\right)=-\int_{-\infty }^{\infty }f_{X}\left(x\right)\log\left[f_{X}\left(x\right)\Delta_{X}\right]dx$ and the corresponding weighted entropy $h^{w}\left(X\right)=-\int_{-\infty }^{\infty }xf_{X}\left(x\right)\log\left[f_{X}\left(x\right)\Delta_{X}\right]dx$. At first glance this appears hopeful since the logarithm arguments are now dimensionless, but let us consider a specific example: the exponential distribution $f_{X}\left(x\right)=\lambda e^{-\lambda t}\;\left(t\geq 0\right)$ where the mean outcome μ=1/λ. This yields $h\left(X\right)=1+\log\left(\mu /\Delta_{X}\right)$ which is (as one would expect) a monotonically increasing function of μ tending to −∞ as μ→0.

However, the weighted entropy $h^{w}\left(X\right)=\mu \left[2+\log\left(\mu /\Delta_{X}\right)\right]$ which experiences a finite minimum when $\mu =e^{-3}\Delta_{X}$. Though dimensionally valid, this creates a dependence on the unit-system used. Figure 1 shows the entropy values plotted against the expectation for calculation in seconds and minutes, showing the shift in the minimum weighted entropy between the two unit systems. The absurdity of this becomes apparent when one considers two exponentially distributed random variables X and Y with E[X] = 9 s and E[Y] = 15 s: Table 1 shows that $h^{w}\left(X\right)>h^{w}\left(Y\right)$ when computed in nat-hours but $h^{w}\left(X\right)<h^{w}\left(Y\right)$ when computed in nat-seconds.

The underlying problem is as follows: since logarithm polarity depends on whether or not $f_{X}\left(x\right)$ exceeds $1/\Delta_{X}$, different sections of the PDF may exert opposing influences on the integral (Figure 2). While this is unimportant for h(X) which has no finite minimum, $h^{w}\left(X\right)$ is forced towards zero with decreasing E[X], which ultimately counteracts the negative-going influence of the logarithm. The two factors therefore operate contrarily: zero surprisal appears as entropy minus infinity and zero importance as entropy zero. Two solutions suggest themselves: (i) combine X and $-\log\left[f_{X}\left(X\right)\Delta_{X}\right]$ in an expression to which they both always contribute positively (e.g., a weighted sum, which in fact yields a weighed sum of expectation and unweighted entropy) and (ii) retain the product but redefine the logarithm argument such that surprisal is always positive. With this in mind, the following section considers the fundamental relationship between absolute and differential entropies.

## 4. Granularity

All physical quantities are ultimately quantified by discrete units; time for example as a number of regularly-occurring events (e.g., quartz oscillations) between two occurrences, which is ultimately limited by the Planck time ($\approx 10^{-43}$ s), though the smallest temporal resolution ever achieved is around $10^{-21}$ s [22]. Finite granularity therefore exists in all practical measurements: if the smallest incremental step for a given system is δx then $f_{X}\left(x\right)$ is really an approximation of a discrete distribution, outcomes 0, δx, 2δx…. having probabilities $f_{X}\left(0\right)\delta x$, $f_{X}\left(\delta x\right)\delta x$, $f_{X}\left(2\delta x\right)\delta x$…etc., so
$$H_{\delta x}\left(X\right)=-\sum_{i=0}^{\infty }f_{X}\left(i\delta x\right)\log\left[f_{X}\left(i\delta x\right)\delta x\right]\delta x$$
which may be expanded into two terms (in the manner of [18])
$$H_{\delta x}\left(X\right)=-\sum_{i=0}^{\infty }f_{X}\left(i\delta x\right)\log\left[f_{X}\left(i\delta x\right)\Delta_{X}\right]\delta x+\log\frac{\Delta_{X}}{\delta x}$$
and if δx is sufficiently small
(8)$$H_{\delta x}\left(X\right)\approx -\int_{0}^{\infty }f_{X}\left(x\right)\log\left[f_{X}\left(x\right)\Delta_{X}\right]dx+\log\frac{\Delta_{X}}{\delta x}=h\left(X\right)+\chi_{X}$$
where the logarithm argument in h(X) is “undimensionalised” (as per Matta et al. [17]) and $\chi_{X}=\log\frac{\Delta_{X}}{\delta x}$ is the information (in nats) needed to represent one dimensioned base-unit in the chosen measurement system: this provides the correctional “shift” needed when the unit-system is changed and thus makes (8) comply exactly with the Π-theorem.

The corresponding weighted entropy may be dealt with in the same manner
$$\begin{array}{ll} H_{\delta x}^{w}\left(X\right) & =-\sum_{i=0}^{\infty }i\delta xf_{X}\left(i\delta x\right)\log\left[f_{X}\left(i\delta x\right)\right]\delta x\\ & \;=-\sum_{i=0}^{\infty }i\delta xf_{X}\left(i\delta x\right)\log\left[f_{X}\left(i\delta x\right)\Delta x\right]\delta x+\log\frac{\Delta_{X}}{\delta x}\cdot \sum_{i=0}^{\infty }i\delta xf_{X}\left(i\delta x\right)\delta x \end{array}$$
(9)$$∴H_{\delta x}^{w}\left(X\right)\approx -\int_{0}^{\infty }xf_{X}\left(x\right)\log\left[f_{X}\left(x\right)\Delta_{X}\right]dx+\log\frac{\Delta_{X}}{\delta x}\cdot E\left[X\right]=h^{w}\left(X\right)+E\left[X\right]\cdot \chi_{X}.$$
While the second term in (9) corresponds to the enigmatic E[X]log(second) “dimension” of (7), it now has an interpretation independent of the measurement system and allows weighted entropies from different distributions to be compared. However, a suitable δx must be chosen; while this need not correspond to the *actual* measurement resolution, it is necessary (in order for all entropy contributions to be non-negative) that $f_{X_{i}}\left(x\right)\cdot \delta x\leq 1$ across all random variables $X_{1}$, $X_{2}$…$X_{N}$ whose weighted entropies are to be compared. It must therefore not exceed
(10)$$\delta x_{max}=1/\underset{i=1,\ldots ,N}{\max}\underset{0\leq x<\infty }{\max}f_{X_{i}}\left(x\right).$$
Similarly, the residual weighted entropy can be shown to be
(11)$$H_{\delta x}^{w}\left(X;t\right)\approx h^{w}\left(X;t\right)+E\left[X|X>t\right]\cdot \chi_{X}$$
where E[X|X>t] is the expectation of X given X>t. The maximum granularity now becomes
(12)$$\delta x_{max}=1/\underset{i=1,\ldots ,N}{\max}\underset{t_{i}\leq x<\infty }{\max}\left[f_{X_{i}}\left(x\right)/{\bar{F}}_{X_{i}}\left(t_{i}\right)\right]$$
where $t_{i}$ is the t-value pertinent to the random variable $X_{i}$ and ${\bar{F}}_{X_{i}}\left(t_{i}\right)$ is the corresponding survival function. Equations (9) and (11) also provide a clue as to the lower acceptable limit of the granularity: if δx were too small then the second terms in these expressions would dominate, making “weighted entropy” merely an overelaborate measure of expectation. Within this window of acceptable values, a compromise “working granularity” must be found. This will be addressed later.

## 5. Gamma, Exponential and Gaussian Distributions

For the purpose of studying this granular entropy, the following specific probability distributions were chosen:
**Exponential**: this is the distribution of time intervals between independent spontaneous events. **Gamma**: this generalizes the Erlang distribution of a sequence of k consecutive identically distributed independent spontaneous events; this generalization allows k to be a non-integer.**Gaussian (Normal)**: this represents the aggregate of many independent random variables in accordance with the central limit theorem. It is also the limit of the gamma as k→∞ and has the largest possible entropy for a given variance [1].

Figure 3 compares examples of the three distributions with the same mean, showing how the exponential and Gaussian are the limiting cases for the gamma distribution for k equal to 1 and infinity respectively. As before, we assume that X represents a time interval (though it could represent other physical quantities).

### 5.1. The Exponential Distribution

The exponential distribution models spontaneous events such as the decay of atoms in a radioactive isotope or “soft” electronic component failures. The PDF is $f_{X}\left(x\right)=\lambda e^{-\lambda t};x\geq 0$ where λ is the “rate parameter”: it has the property that 1/λ is both the expectation *and* the standard deviation. Applying (9) and (11) we find that the regular and weighted entropies are:(13)$$H_{\delta x}\left(X\right)=1-\log\left(\lambda \delta x\right),$$
(14)$$H_{\delta x}^{w}\left(X\right)=\frac{1}{\lambda }\left[2-\log\left(\lambda \delta x\right)\right].$$
Residual weighted entropy is worked out as an example in [13] (p. 9): “granularized”, it can be written
(15)$$H_{\delta x}^{w}\left(X;t\right)=t+\frac{2}{\lambda }-\left(t+\frac{1}{\lambda }\right)\log\left(\lambda \delta x\right)$$
which is clearly a linear function of t with gradient 1−log(λδx). In the original formulation (with $\Delta_{X}$ in place of δx) this was problematic since the slope could be either positive and negative, but now by keeping λδx≤1 we ensure the weighted entropy never decreases with t and always remains constant or decreases with increasing λ.

### 5.2. The Gamma Distribution

The PDF of the gamma distribution is:(16)$$f_{X}\left(x\right)=\frac{\lambda }{\Gamma \left(k\right)}{\left(\lambda x\right)}^{k-1}e^{-\lambda x};x\geq 0$$
where $\Gamma \left(k\right)=\int_{0}^{\infty }z^{k-1}e^{-z}dz$ (the gamma function). Since the variance $\sigma^{2}=k\times 1/\lambda^{2}$ and the expectation E[X]=k/λ, we can obtain a gamma distribution with any desired expectation and standard deviation by setting $k=E{\left[X\right]}^{2}/\sigma^{2}$ and $\lambda =E\left[X\right]/\sigma^{2}$. Substituting (16) into (8) yields
(17)$$H_{\delta x}\left(X\right)=k-\left(k-1\right)\psi \left(k\right)+\log\Gamma \left(k\right)-\log\left(\lambda \delta x\right)$$
where ψ(x)=dlogΓ(k)/dk (the digamma function).

Since Γ(1)=1, (17) simplifies to (13) when k=1. Similarly, the weighted entropy can be written
(18)$$H_{\delta x}^{w}\left(X\right)=\frac{1}{\lambda }\left[k+1-\left(k-1\right)\psi \left(k+1\right)+\log\Gamma \left(k\right)-\log\left(\lambda \delta x\right)\right].$$
Using the recursive property ψ(k+1)=ψ(k)+1/k [23] and recalling that E(X)=k/λ, we uncover a very simple relationship between the weighted and unweighted entropies:(19)$$H_{\delta x}^{w}\left(X\right)=\frac{1}{\lambda }\left[1+kH_{\delta x}\left(X\right)\right]=\frac{1}{\lambda }+E\left(X\right)H_{\delta x}\left(X\right).$$
(Note that (18) and (19) are consistent with (13) and (14) for k=1.) In a similar manner, the residual weighted entropy can be shown to be
(20)$$H_{\delta x}^{w}\left(X;t\right)=\frac{1}{\lambda \Gamma \left(k,\lambda t\right)}\left[\Gamma \left(k+1,\lambda t\right)\log\frac{\lambda \delta x}{\Gamma \left(k,\lambda t\right)}+\left(k-1\right)\Lambda \left(k,\lambda t\right)-\Gamma \left(k+2,\lambda t\right)\right]$$
where $\Gamma \left(y,z\right)=\int_{z}^{\infty }x^{y-1}e^{-x}dx$ (the upper incomplete gamma function) and $\Lambda \left(y,z\right)=\int_{z}^{\infty }x^{y}e^{-x}\log xdx$. Though not a well-recognized function, this converges for all y,z>0, may be defined 0 for z=0 (y>0) and computed to any required degree of accuracy using Simpson’s rule. Also note that when k=1, the term containing Λ vanishes and (20) simplifies to (15).

### 5.3. The Gaussian (or Normal) Distribution

The PDF of the Gaussian distribution is given by
(21)$$f_{X}\left(x\right)=\frac{1}{\sigma \sqrt{2\pi }}\exp\left[-\frac{1}{2}{\left(\frac{x-\mu }{\sigma }\right)}^{2}\right];\;-\infty <x<\infty$$
where μ is the expectation and σ the standard deviation. While the distribution extends to infinity in both directions (unlike the exponential and gamma which are defined only for x≥0) we have been considering temporal separation which can only be positive; for this reason we impose an additional restriction that σ≤μ/3 such that Pr(X<0) never exceeds 0.0013, which may, for practical purposes, be neglected. The expression for the Shannon differential entropy has already been introduced in Section 2; “granularized”, the expression may be written
(22)$$H_{\delta x}\left(X\right)=\log\sqrt{2\pi }+\log\frac{\sigma }{\delta x}+\frac{1}{2}.$$
For the weighted entropy we substitute (21) into (9) and simplify to obtain
(23)$$H_{\delta x}^{w}\left(X\right)=\mu \left[\log\sqrt{2\pi }+\log\frac{\sigma }{\delta x}+\frac{1}{2}\right]=E\left[X\right]H_{\delta x}\left(X\right).$$
So, the weighted entropy is simply the unweighted entropy multiplied by the expectation. With the exception of the 1/λ term this is almost the same as (19), and as k becomes large the two expressions converge. This is to be expected since the central limit theorem [24] requires that the sum of many independent random variables behaves as a Gaussian: since the gamma distribution represents the convolution of k exponentials (each with expectation 1/λ), when k is large (and thus 1/λ small) the gamma and Gaussian acquire near-identical properties for x>0.

To obtain an expression for the residual weighted entropy of the Gaussian distribution we substitute (21) into (11) and simplify to obtain:(24)$$H_{\delta x}^{w}\left(X;t\right)=\frac{1}{\sqrt{\pi }}\left[\mu a-\sigma \sqrt{2}\left(b-a^{2}-1\right)\right]\frac{e^{-a^{2}}}{\mathrm{erfc}\left(a\right)}-\mu \left[b-\frac{1}{2}\right]$$
where $a=\frac{t-\mu }{\sigma \sqrt{2}}$, $b=\log\frac{\delta x\sqrt{2/\pi }}{\sigma \mathrm{erfc}\left(a\right)}$ and $\mathrm{erfc}\left(a\right)=\frac{2}{\sqrt{\pi }}\int_{a}^{\infty }e^{-z^{2}}dz$ (the complementary error function).

### 5.4. Numerical Calculations

Figure 4 shows the weighted entropies computed for exponential, Gaussian and gamma distributions for a mean outcome of 1 s and two levels of granularity (0.05 and 0.1 s) across a range of standard deviations less than the mean. We make the following observations:
Both the weighted and unweighted entropies for σ=0 should in principle be zero (since here $f_{X}\left(x\right)$ becomes a Dirac delta function located at x=μ) but would actually tend to minus infinity as σ→0. Our “granularized” entropy definitions (8) and (9) cease to be meaningful in this region since they approximate absolute entropies which must be non-negative.Meaningful Gaussian curves cannot be computed for σ≳0.3 since this would significantly violate the assumption that all X>0 (see Section 5.3). Thus, the gamma “takes over” from the Gaussian across the range 0.3≲σ≲1.0, thus providing a kind of “bridge” to the exponential case on the far right.Although $H_{\delta x}\left(X\right)$ ceases to rise significantly beyond σ≈0.8, $H_{\delta x}^{w}\left(X\right)$ increases almost linearly up to σ=1.0. This is because the expanding upper tail of the distribution, though not significantly increasing the surprisal, nevertheless causes larger X values to contribute more significantly.

## 6. Choosing a Working Granularity

In Section 4 we postulated the existence of a “window” from which an acceptable working value of δx must be chosen. Though we did not specify its limits, we noted that if δx were too small it would eliminate the nuance of “entropy” from $H_{\delta x}^{w}\left(X;t\right)$ and make it merely an overelaborate measure of expectation. For this reason, we suggest that δx be as large as possible, though not so large as to exceed the reciprocal of the maximum probability density and thus introduce negative surprisal. Here we test this suggestion and explore its implications for the distributions previously described.

### 6.1. The Upper Limit

The exponential distribution has the property $f_{X}\left(x\right)/\bar{F}\left(t\right)=f\left(x-t\right)$ of which the maximum is always constant and equal to the rate parameter λ. Thus if all distributions to be compared are exponential then $\delta x_{max}=1/\underset{1\leq i\leq N}{\max}\lambda_{i}$ where $\lambda_{i}$ is the rate parameter for $f_{X_{i}}\left(x\right)$ independent of t. However, this property does not apply to the more general gamma distribution, whose modal value (k−1)/λ when substituted into (16) and (12) (noting that $\bar{F}\left(t\right)=\Gamma \left(k,\lambda t\right)/\Gamma \left(k\right)$) gives
(25)$$\underset{t\leq x<\infty }{\max}\frac{f_{X}\left(x\right)}{{\bar{F}}_{X}\left(t\right)}=\{\begin{array}{cc} \frac{\lambda }{\Gamma \left(k,\lambda t\right)}{\left(k-1\right)}^{k-1}e^{-\left(k-1\right)}; & t\leq \left(k-1\right)/\lambda \\ \frac{\lambda }{\Gamma \left(k,\lambda t\right)}{\left(\lambda t\right)}^{k-1}e^{-\lambda t}; & t>\left(k-1\right)/\lambda \end{array}.$$
Similarly for the Gaussian distribution the overall maximum probability density $1/\sigma \sqrt{2\pi }$ occurs when x=μ and $\bar{F}\left(t\right)=\frac{1}{2}\mathrm{erfc}\left(\frac{t-\mu }{\sigma \sqrt{2}}\right)$ so the maximum value for the range t≤x<∞ must be
(26)$$\underset{t\leq x<\infty }{\max}\frac{f_{X}\left(x\right)}{{\bar{F}}_{X}\left(t\right)}=\{\begin{array}{cc} \frac{\sqrt{2/\pi }}{\sigma \cdot \mathrm{erfc}\left(\frac{t-\mu }{\sigma \sqrt{2}}\right)}; & t\leq \mu \\ \frac{\sqrt{2/\pi }}{\sigma \cdot \mathrm{erfc}\left(\frac{t-\mu }{\sigma \sqrt{2}}\right)}\exp\left[-\frac{1}{2}{\left(\frac{t-\mu }{\sigma }\right)}^{2}\right]; & t>\mu \end{array}$$
Calculations were performed on a set of four distributions, each with an expected value of 1.0 s:
Gaussian with standard deviation 0.3 sGamma with k=5 (standard deviation 0.447 s) Gamma with k=15 (standard deviation 0.258 s)Exponential with λ=1.0 s^−1^. 

Figure 5 shows the residual PDFs for the first of these with the maximum probability density overlaid. Figure 6 compares this with the other three distributions: the maximum for the entire set (for 0≤t≤1.5 s) is 6.938 s^−1^, so from (12) the maximum allowable granularity for comparing their entropies $\delta x_{max}$ = 1/6.938 = 0.144 s. 

While this ensures positive surprisal throughout our range of interest, the granularity may nevertheless be subject to other constraints. To investigate further we introduce an alternative calculation for weighted entropy to which (9) and (11) may be compared. Consider a “histogram” of C cells, each δx wide, the cell i∈{1,…,C} having constant probability $P\left(i\right)=\int_{x_{i}}^{x_{i}+\delta x}f_{X}\left(x\right)dx$ ($x_{i}$ being the lower cell boundary and $x_{1}=t$). Figure 7 shows the histograms for the Gaussian distribution with three different granularities (C taking the minimum value required to cover the range [0, 6σ]) and t=0. Clearly as δx increases the discretized distribution resembles less the corresponding continuous distribution. In each case the weighted entropy can be approximated
(27)$${\bar{H}}_{\delta x}^{w}=-\overset{C}{\underset{i=t/\delta x}{\sum }}{\bar{x}}_{i}\frac{P\left(i\right)}{\bar{F}\left(t\right)}\log\frac{P\left(i\right)}{\bar{F}\left(t\right)}\delta x$$
where $\bar{x_{i}}$ is the horizontal position of the centroid of the PDF enclosed by cell i and $\bar{F}\left(t\right)=1-\sum_{0}^{i=t/\delta x}P\left(i\right)$.

Figure 8 compares the results of (27) with those of (24) across a range of granularities for t=0, showing the values are almost identical for small δx but diverge as the granularity increases. The “upper limit” $\delta x=\underset{t\leq x<\infty }{max}f_{X}\left(x\right)/\bar{F}\left(t\right)$ = 0.752 s (represented by the three-cell distribution in Figure 7) shown by the broken line appears to represent the lower boundary for large errors, though noticeable discrepancies do exist for all δx greater than half this value.

We therefore define the upper limit of granularity as the maximum δx for which the two weighted entropy approximations disagree by no more than a fraction α of their combined average, i.e.,
(28)$$\left|{\bar{H}}_{\delta x}^{w}\left(X;t\right)-H_{\delta x}^{w}\left(X;t\right)\right|=\alpha \frac{{\bar{H}}_{\delta x}^{w}\left(X;t\right)+H_{\delta x}^{w}\left(X;t\right)}{2}$$
which may be computed iteratively for any given distribution and t-value. We choose as our benchmark α = 0.321 (i.e., 32.1% maximum error) which corresponds to the previously computed δx = 0.752 s and plot the upper limits of δx for a range of σ values (see Figure 9).

The granularity computed from (12) is mostly lower (and never significantly higher) than the value from (28) and the former could be regarded as a cautious “engineering” lower limit: for the range of distributions compared in Figure 6 this is 0.144s and Figure 10 shows the weighted entropy calculated using this value across the same range of t. We make the following observations:
The residual weighted entropy for the exponential distribution has the strongest dependence on t; since the distribution shape (and variance) does not depend on t, the increase in weighted entropy is caused entirely by the increased mean weighting.For the gamma distribution with k=5, the residual variance decreases with increasing t, the distribution becoming progressively more concentrated around its mean. This causes the entropy to fall, counteracting somewhat the increased weightings. Thus, the rise in weighted entropy with increasing t is less pronounced than for the exponential distribution.For the gamma distribution with k=15, these competing effects almost cancel each other, the decreased variance compensating almost exactly for the increased average weighting.The Gaussian results are similar, the weighted entropy now showing a pronounced decrease with increasing t. Re-plotting the Gaussian graph for smaller δx-values (Figure 11) shows that a critical granularity exists (in this case δx = 0.0465 s) where the residual weighted entropy remains almost constant as t is varied.

### 6.2. The Lower Limit

Having established an upper limit for granularity, we observe the effect of using lower values than this. Figure 11 shows results obtained from the Gaussian PDF, indicating that with different granularities the weighted entropy can both rise and fall with increasing t, a situation not unlike that which arose from applying Molyneaux’s dimensionality approach to differential weighted entropy (see Section 3): the largest weighted entropy amongst a group of distributions now depends not on measurement units but on the measurement granularity. This is to be expected since as δx decreases, $H_{\delta x}^{w}\left(X;t\right)$ becomes progressively more “expectation-like” due to the increased influence of the second term in (11). However, we must ask at what point does $H_{\delta x}^{w}\left(X;t\right)$ cease to be a meaningful “entropy” and merely a measure of expectation? What additional condition might be imposed to prevent this from happening?

One possibility would be to constrain the granularity such that two scenarios, one with higher and the other with lower entropy should never be allowed to switch over when granularity is changed. However, there remains the possibility that acceptable δx ranges for different distributions to be compared do not overlap, and some may have to be compared with others based solely on an expectation-like weighted entropy.

## 7. Conclusions

We have identified and attempted to address the dimensionality problem present in Di Crescendo and Longobardi’s differential residual weighted entropy formulation [13]—namely the opposing influences of the positive and negative values of $\log f_{X}\left(x\right)$ which (since $f_{X}\left(x\right)$ is a dimensioned quantity) depend on the unit system. This does not affect Shannon’s differential entropy [1] so long as consistent units are employed, but it does become important when x appears as an all-positive weighting. We circumvent this problem by applying a “working granularity” δx to convert differential entropy into a “quasi-absolute” quantity, choosing δx to be the largest value required to make $\log f_{X}\left(x\right)\delta x\leq 0$ in all distributions of interest. We demonstrate this formulation using the residual exponential, gamma, and Gaussian distributions. There are many other issues to be investigated: firstly, we have assumed throughout a single random variable X whose sample values are uncorrelated. The extension of this idea to the strongly correlated Tsallis [9,10] entropy definition remains to be explored. Furthermore, the application to joint entropies in multivariate distributions has yet to be investigated.

## Figures and Tables

**Figure 1 entropy-21-00825-f001:**
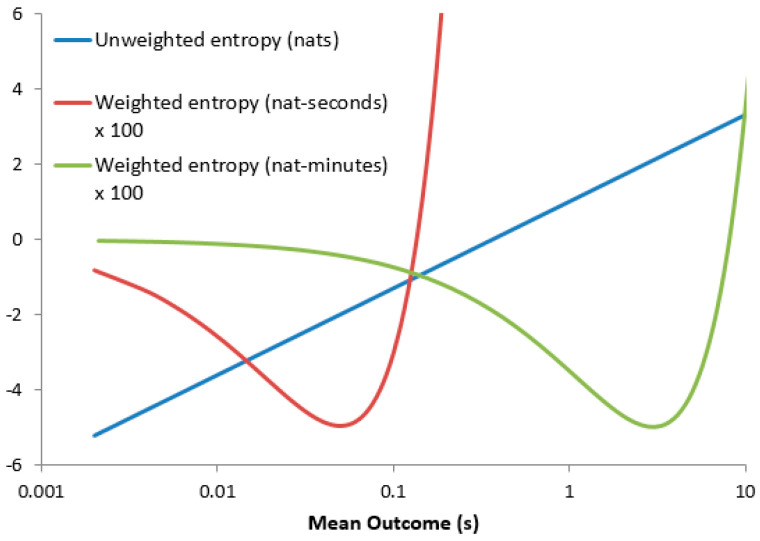
Weighted and unweighted differential entropies for an exponential distribution plotted against expected outcome, calculation performed in seconds and minutes.

**Figure 2 entropy-21-00825-f002:**
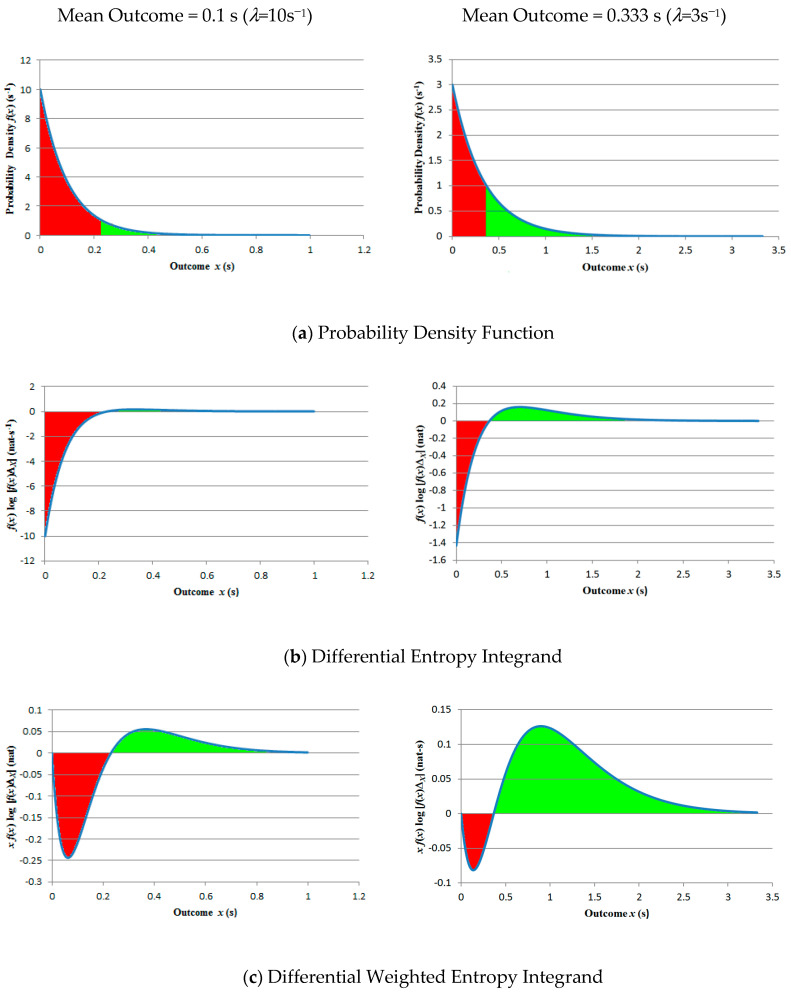
Positive and negative contributions of probability density to the unweighted and weighted entropies for two exponential distributions: (**a**) shows the PDFs, (**b**) the unweighted and (**c**) the weighted entropies. Positive contributions are shown in green and negative contributions in red.

**Figure 3 entropy-21-00825-f003:**
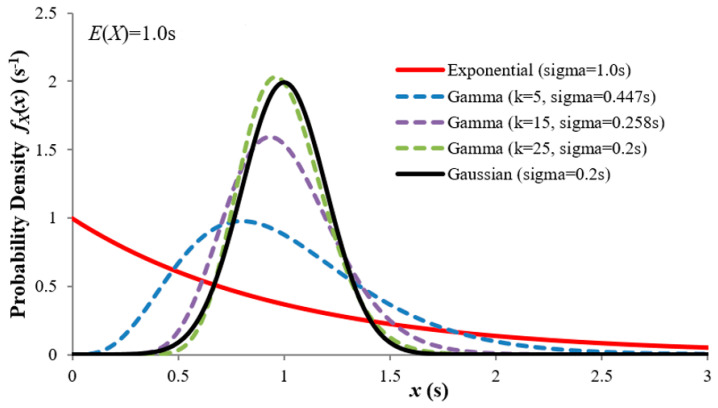
Exponential, gamma and Gaussian distributions. Gamma for k=1 is identical to the exponential, while the Gaussian is the limiting case of gamma as k→∞.
E(X)=k/λ=1 for all the gammas, and the Gaussian σ=0.2s (that of the gamma for k=25).

**Figure 4 entropy-21-00825-f004:**
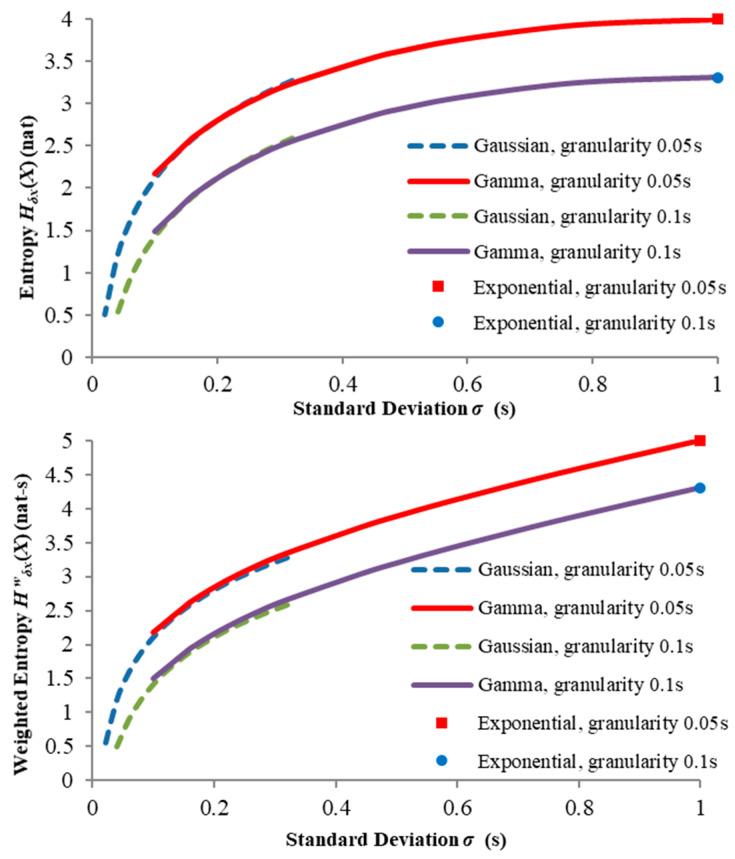
Gamma, Gaussian and exponential entropies for two different granularities.

**Figure 5 entropy-21-00825-f005:**
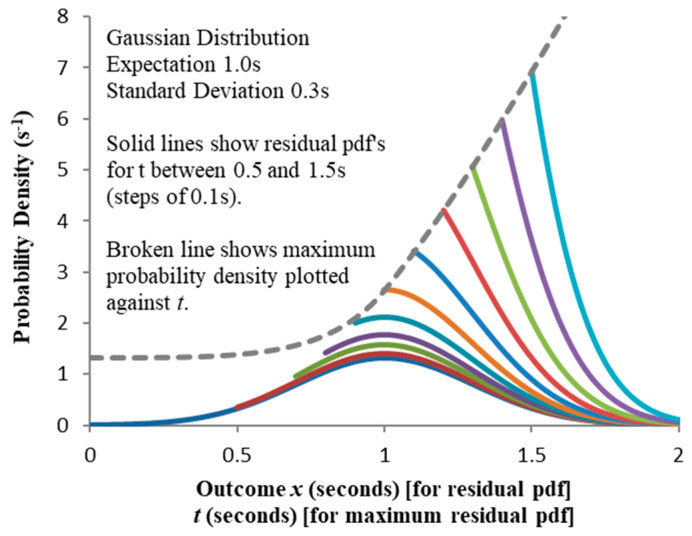
Residual probability density functions for the Gaussian distribution; the maximum probability density is overlaid as a function of t.

**Figure 6 entropy-21-00825-f006:**
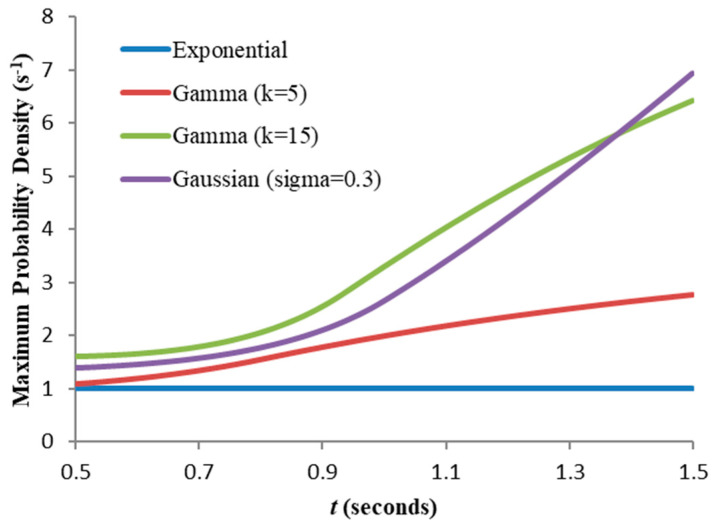
Comparison of maximum residual probability densities for four distributions with the same mean (1.0 s). The observed boundary maximum of the probability density for this range is 6.938s^−1^.

**Figure 7 entropy-21-00825-f007:**
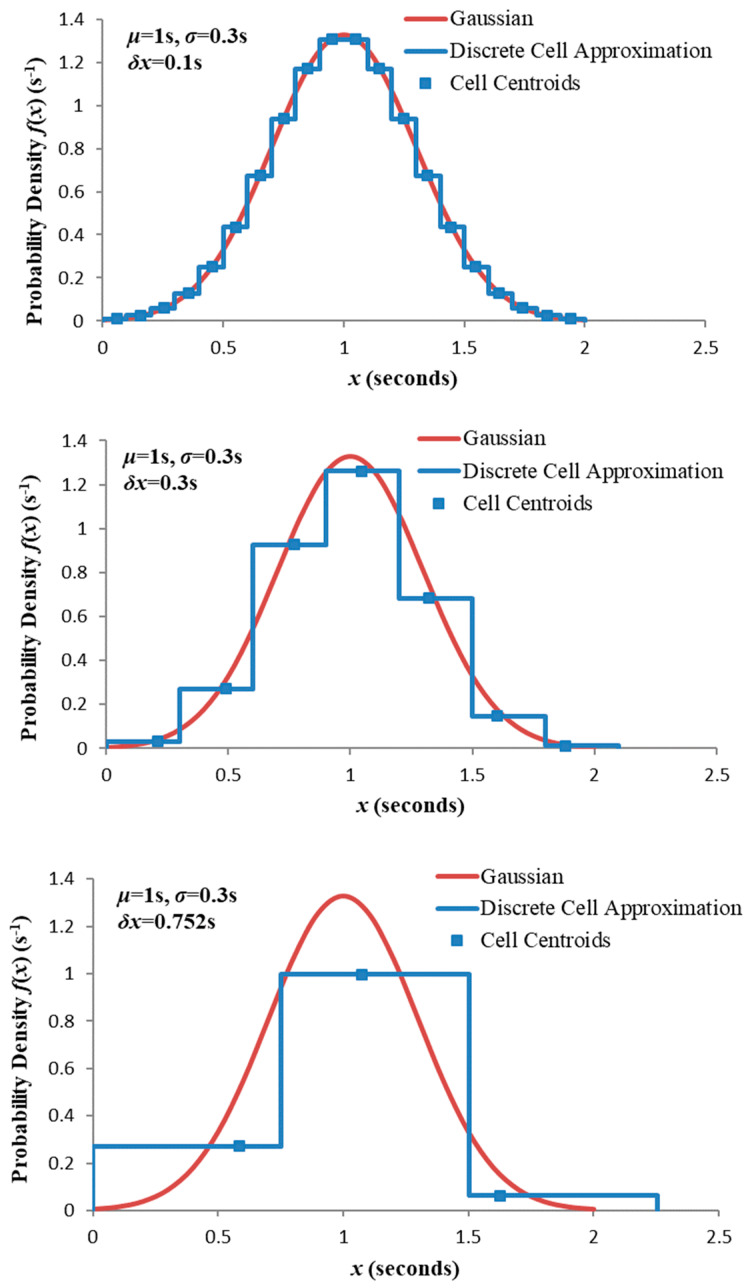
Comparison of Gaussian PDF and discrete “histogram” approximation for three granularities.

**Figure 8 entropy-21-00825-f008:**
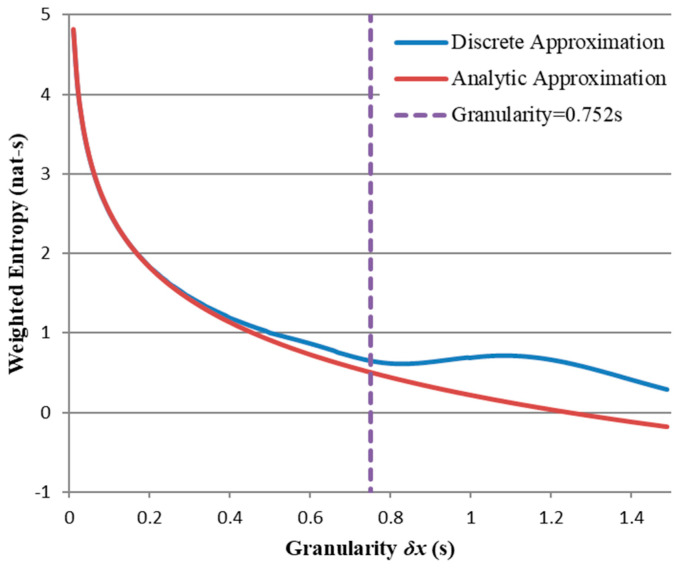
Weighted entropies for Gaussian μ=1 s, σ=0.3 s, computed using (23) and (27). Broken line indicates $\delta x=\underset{t\leq x\leq \infty }{\max}f_{X}\left(x\right)/\bar{F}\left(t\right)$ = 0.752.

**Figure 9 entropy-21-00825-f009:**
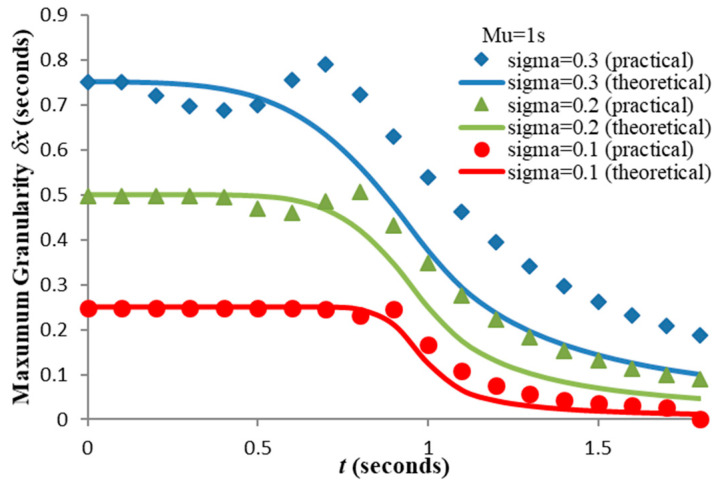
Upper granularity limits for 32.1% error between “theoretical” (12) and “practical” (28) residual weighted entropies for Gaussian (μ = 1.0 s, σ = 0.1, 0.2, 0.3 s).

**Figure 10 entropy-21-00825-f010:**
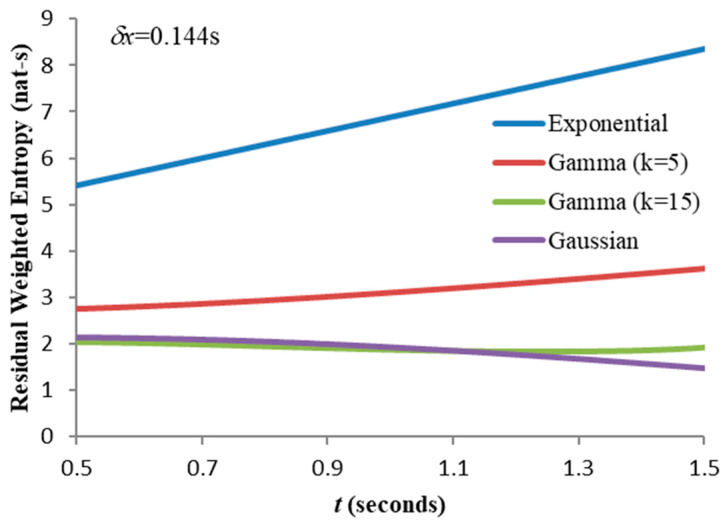
Comparison of residual weighted entropies for the distributions of Figure 6 (expectations again set to 1.0).

**Figure 11 entropy-21-00825-f011:**
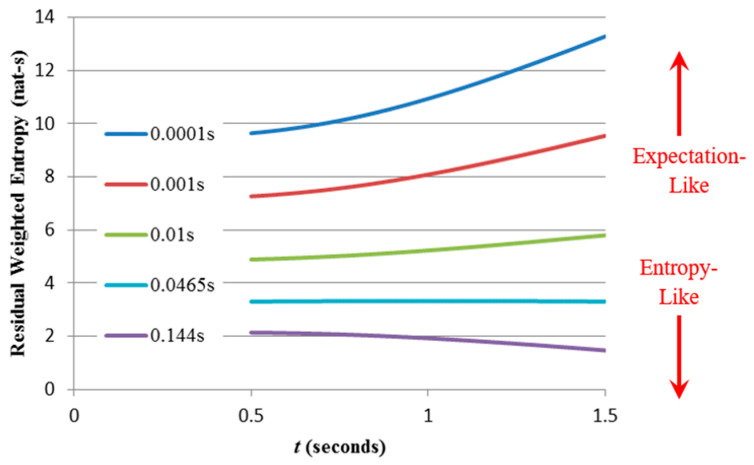
The effects of mean and variance on residual weighted entropy of a Gaussian RV (μ = 1 s, σ = 0.3 s) for different values of δx (noted on the left for each curve) which clearly determines the dominating influence. There exists a critical value (δx
≈ 0.0465 s) where the residual weighted entropy barely depends upon t.

**Table 1 entropy-21-00825-t001:** Comparison of the weighted and unweighted entropies for two exponential processes. Entropy units are nats (unweighted) and nat-seconds/nat-hours (weighted).

	E(*X*) = 9 s (0.0025 h)		E(*Y*) = 15 s (0.00417 h)		Entropy Increase	
Measurement Units	Seconds	Hours	Seconds	Hours	Seconds	Hours
Unweighted Entropy	3.1972	−4.9915	3.7081	−4.4806	0.511	0.511
Weighted Entropy	37.775	−0.01	70.621	−0.0145	32.85	−0.0045

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
