# Peer review of "Dimensionality, Granularity, and Differential Residual Weighted Entropy"

_entropy, 2019, doi:10.3390/e21090825_

Round 1

Reviewer 1 Report

The article deals with issue of the transition from the continuous to the discrete, always an important issue. It does so in the framework of Shannon entropy, which is primarily defined in terms of discrete distributions and then generalized to continuous ones. The authors show some problems that occur in the continuous version of the entropy, and offer to solve them by discretizing some of the continuous distributions. The article is written with the aim to be didactic, and I think it achieves that. 

In section 4, on granularity, the author's claim that the world is not continuous is backed up by recent research on quantum theories of gravity, which show that space at the Planck level is granular, and not continuous. Also, a continuous space would lead to the possibility of concentrating an infinite amount of information in an infinitesimally small space, what contradicts the principles of quantum mechanics and is likely to be one of the main issues leading to problems with continuous formulations of entropy, that measures information.

It seems to me the authors use a particular weight function in order to study weighted entropies, and that all results depend on that particular choice, although it is not clearly specified in the article.

So, I don't have any objections for the article to be published in Entropy.

The orthography and grammar are quite good, but I spotted some missing comas, as follows:

Line 133 "specific distribution,";

Line 187 "(7), it";

Line 201: "entropy, the";

Line 222: ""granularized", it";

Line 230: ", we can";

Line 234: ", we uncover";

Line 236: "manner, the".

Author Response

We are grateful for this reviewer's comments. These are our responses to particular points raised:

It seems to me the authors use a particular weight function in order to study weighted entropies, and that all results depend on that particular choice, although it is not clearly specified in the article.

The idea of a "weight function" was introduced by GuiaÅŸu but  subsequent workers (notably Di Crescenzo and Longobardi) simplified matters by using the r.v. itself as a "weight". Nonlinear weight functions could still be accommodated in this scheme by redefining the r.v. as its mapping through the function (though this would of course depend on the latter being monotonic).

The orthography and grammar are quite good, but I spotted some missing comas, as follows:

Line 133 "specific distribution,";

Line 187 "(7), it";

Line 201: "entropy, the";

Line 222: ""granularized", it";

Line 230: ", we can";

Line 234: ", we uncover";

Line 236: "manner, the".

We thank the reviewer for pointing out these errors, and have made the necessary corrections.

Reviewer 2 Report

This paper tries to address the dimensionality problem of the entropy formulation and provides sufficient evidence for their findings. My concern is this paper may not be friendly enough to all the potential readers of the entropy journal, it is better to introduce why this issue is critical at the beginning with plain language. 

Author Response

We are grateful to the reviewer for these comments. Here is our response to the issue raised:

"My concern is this paper may not be friendly enough to all the potential readers of the entropy journal, it is better to introduce why this issue is critical at the beginning with plain language."

The final paragraph of the introduction has been expanded, to give more emphasis to the importance of the topic.

Reviewer 3 Report

The paper addresses a known problem with the weighted entropy of dimensioned quantities: the result depends on the choice of units (seconds, minutes, days, etc.). The suggestion presented in the paper is not to use arbitrary man-made units, but rather a special unit, called "working granularity" in this paper, chosen according to a maximizing principle.

Overall, the paper is well presented and sound, and I do recommend publishing it. I have one major comment, and several minor comments:

1. The maximum observed probability density is measured for a set of specific probability distributions, but if a different probability distribution is chosen it might be higher. Is there a way to find the maximum for arbitrary PDFs, perhaps using calculus of variations? In other words, the working granularity is set for each group of PDFs but might be different for another group. Is there a way to find a working granularity which works for all types of PDFs?

2. Please provide references for the claims in lines 56-58. Specifically "h(x) is not the limit of H(x)", "h(x) may be negative" and "entropy for deterministic outcome is not zero".

3. I think there is a typo in Eq. (3). The argument of f_X should be lower case x, not upper case X.

4. In several places the weighted functions h^w(x) and H^w(X) are written without ^w. Please fix this recurring typo.

5. Typo in figure 6: the maximum probability density is 6.938 not 0.938. Also, this maximum is only true for this chosen time window.

Author Response

We thank the reviewers for these comments. Here are our responses to the particular issues raised:

1. The maximum observed probability density is measured for a set of specific probability distributions, but if a different probability distribution is chosen it might be higher. Is there a way to find the maximum for arbitrary PDFs, perhaps using calculus of variations? In other words, the working granularity is set for each group of PDFs but might be different for another group. Is there a way to find a working granularity which works for all types of PDFs?

We agree that this is an important issue: in order to compare a group of p.d.f.'s, one must know in advance the maximum probability density within that group. (Just as to calibrate a thermometer one must know in advance the highest and lowest temperatures to be measurable.) The idea of using calculus of variations to determine the maximum value of an arbitrary p.d.f. is interesting, but not something that can be realistically investigated within the time frame allowed for "minor revisions". However, we are inclined to think that the largest possible value for the p.d.f. must be the reciprocal of the measurement uncertainty, which would make the working granularity equal to that uncertainty. Since this is likely to be very small, it would make the E(X).chi_X term dominate, and hence make the "weighted entropy" merely a measure of expectation. This is one thing we were trying to avoid. However, this idea is certainly worth some more thought and we are grateful to the reviewer for suggesting it.

2. Please provide references for the claims in lines 56-58. Specifically "h(x) is not the limit of H(x)", "h(x) may be negative" and "entropy for deterministic outcome is not zero".

We have included a reference to a theorem in the Cover and Thomas book about the limiting value of discrete entropy. The potential negativity of h(X) results when f(x)>1 across a significant range of the support and we have reworded the text to make this clearer to the reader. We further invite the reader to consider a deterministic outcome as a limiting case of this; as the p.d.f. approaches a delta function, log(f(x)) tends to infinity at the point of certainty. The effect on entropy can easily be demonstrated by (i) choosing e.g. a uniform rectangular p.d.f., working out h(X) and finding its limit as the function width tends to zero, or (ii) using the property that the integral of f(x).delta(x-a) dx (across any range including a) = f(a). We hope this is clearer now - though we will add more references if the reviewer feels this is necessary.

3. I think there is a typo in Eq. (3). The argument of f_X should be lower case x, not upper case X.

4. In several places the weighted functions h^w(x) and H^w(X) are written without ^w. Please fix this recurring typo.

5. Typo in figure 6: the maximum probability density is 6.938 not 0.938.

We have corrected these errors, and thank the reviewer for pointing them out.

Also, this maximum is only true for this chosen time window.

Failing to mention that this was a boundary maximum for the range of interest, rather than a local minimum, was quite a serious omission on the part of the authors, and we thank the reviewer for pointing it out. It has now been corrected.